# Study of the Effect of Current Pulse Frequency on Ti-6Al-4V Alloy Coating Formation by Micro Arc Oxidation

**DOI:** 10.3390/ma12233983

**Published:** 2019-12-01

**Authors:** Alexander Sobolev, Alexey Kossenko, Konstantin Borodianskiy

**Affiliations:** Department of Chemical Engineering, Ariel University, Ariel 40700, Israel; sobolev@ariel.ac.il (A.S.); kossenkoa@ariel.ac.il (A.K.)

**Keywords:** micro arc oxidation, current pulse frequency, Ti-6Al-4V alloy, titanium oxide, corrosion resistance

## Abstract

The micro arc oxidation (MAO) process has been applied to produce ceramic oxide coating on Ti-6Al-4V alloy. The MAO process was carried out at the symmetric bipolar square pulse in electrolyte containing Na_2_CO_3_ and Na_2_SiO_3_. The effect of current frequency on the surface morphology, the chemical and the phase compositions as well as the corrosion resistance was examined. Morphology and cross-sectional investigation by electron microscopy evaluated more compacted and less porous coating produced by high current frequency (1000 Hz). This alloy also exhibited a high corrosion resistance in comparison with the untreated alloy. Additionally, the alloy subjected to MAO treatment by a current frequency of 1000 Hz showed a higher corrosion resistance in comparison with alloys obtained by lower current frequencies. This behavior was attributed to more compacted and less porous morphology of the coating.

## 1. Introduction

Titanium alloys are widely applicable in marine, aerospace, chemical and biomedical industries due to their high specific strength, corrosion resistance and excellent biocompatibility [1,2,3,4]. However, their advanced properties are limited due to the lack of the protective surface properties. One of the most promising environmentally friendly processes of coating formation is micro arc oxidation (MAO).

MAO is an electrochemical processing of ceramic oxide coating formation on the metallic substrate by the appearance of dielectric breakdown and plasma discharges. The basic principle of the MAO technology is an application of a high voltage between the treated metal and the electrode. During the process, micro-arc discharge migration points appear on the surface lead to the oxide coating formation [5,6,7,8]. Usually, MAO treatment is suitable for the enhanced mechanical properties and corrosion resistance of the alloys’ surface [9,10,11].

Ti-6Al-4V is one of the most commonly applicable titanium alloys because of its outstanding properties [12]. This Ti alloy contains both aluminum and vanadium; Al stabilizes α phase and has the HCP lattice, and V stabilizes β phase and has the BCC lattice. Consequently, this alloy exhibits advanced mechanical properties as α phase provides a high strength while β phase provides high ductility. Moreover, Ti-6Al-4V alloy is one of the main metallic biomaterials due to its high biocompatibility [13,14,15]. Additional treatment of this alloy by MAO process will lead to the production of an oxide coating with excellent corrosion resistance for applications in wide range of industries.

The majority of research works have been devoted to studying the electrolyte chemical composition effect on the structure and the properties of the created coatings [16,17,18,19]. However, the influence of the process electrical parameters affects also the formation of the coating, resulting in different surface properties [20,21].

Therefore, the goal of the present work is an investigation of the current pulse frequency effect of the MAO process on the coating formation on Ti-6Al-4V alloy. The obtained coating morphology, and the chemical and the phase compositions were also examined in details in the work. Furthermore, the influence of the current frequency on the corrosion resistance of the produced alloys’ surface was also examined.

## 2. Materials and Methods

### 2.1. Coating Production

Ti-6Al-4V alloy samples (Grade V, max. wt.%: 5.5–6.7 Al, 3.5–4.0 V, 0.25 Fe, 0.2 O, 0.05 N, 0.08 C, bal. Ti) (Scope Metals Group Ltd., Israel) with a surface area of 62 cm^2^ were grounded up to abrasive paper grits #1200 and subjected to the ultrasonic cleaning in acetone.

The MAO process was carried out by MP2-AS 35 power supply (Magpulls, Sinzheim, Germany) with the follow electrical parameters: I_max_ = 5 A, U_max_ = 1000 V. Electrical parameters were pulsed (symmetric bipolar square pulse) at a frequency from 200 to 1000 Hz. Names of the experimental samples and their current pulse parameters listed in Table 1. The process was carried out in electrolyte contained sodium carbonate (Na_2_CO_3_, Sigma Aldrich, St. Louis, MO, USA) = 10 g/L and sodium metasilicate pentahydrate (Na_2_SiO_3_·5H_2_O, Sigma Aldrich, St. Louis, MO, USA) = 2 g/L. The process was conducted at 30 °C; in a water-cooled glass vial for 15 min. The examined alloy served as a working electrode and a stainless-steel sheet served as a counter electrode. The anodic current density was set at 0.05 A/cm^2^ and the applied voltage was 250 V, with the duty cycle 50% and symmetrical anode/cathode pulse. Electrical parameters of the process were recorded by Scope Meter 199C, 200 MHz, 2.5 GS s^−1^ (Fluke, Everett, WA, USA).

### 2.2. Coating Characterization

Scanning electron microscope (SEM) MAIA3 (TESCAN, Brno, Czech Republic) was used for surface morphology and cross-sectional examinations. EDS system by (Oxford instruments, Abingdon, UK) with an X-Max^N^ detector was applied for elemental analysis detection. The phase analysis of the coating was detected using the X’Pert Pro diffractometer (PANalytical B.V., Almelo, the Netherlands) with Cu_α_ radiation (λ = 1.542Å) at the grazing incidence mode (grazing angle of 3°) with a 2θ range from 20 to 70° (step size of 0.03°) at 40 kV and 40 mA.

Coating elemental analysis was also examined by particle induced X-ray emission (PIXE) method by 1.7 MV Pelletron accelerator equipped with SuperSilicon Drift Detector, X-123 SDD (Amptek Inc., Bedford, MA, USA) positioned at 45° to the beam (I~4.3 nA with a nominal diameter of 1.5 mm). 2.017 MeV 4He^+^ ± 1 KeV beam was used to collect spectra. The samples were coated by carbon in order to prevent charge effect during irradiation. GUPIX software (University of Guelph, ON, Canada) was used for the PIXE data analysis.

Light scattering measurements were performed using XploRA ONE™ Raman confocal microscope (HORIBA Scientific, Piscataway, NJ, USA) using 532 nm laser excitation line and a constant power of 20 mW.

### 2.3. Coating Corrosion Resistance Test

The corrosion resistance examination was done on PARSTAT 4000A potentiostat/galvanostat (Princeton Applied Research, Oak Ridge, TN, USA) using potentiodynamic polarization test in a 3.5 wt.% NaCl (Sigma-Aldrich, St. Louis, MO, USA) solution, using a three- electrode cell configuration. Pt sheet served as a counter electrode and saturated Ag/AgCl (Metrohm Autolab B.V., Almelo, the Netherlands) served as a reference electrode. The polarization resistance was detected in the range of ± 0.25 V with respect to the recorded corrosion potential at a scan rate of 0.1 mV/s. Prior to the test, the samples were placed in the 3.5 wt.% NaCl solution for 1 h in order to reach the steady state of a working electrode.

## 3. Results and Discussion

### 3.1. Process Characterization

The recorded electric characteristics in the process are illustrated in Figure 1.

Figure 1 presents a typical voltage- and current-time behavior for examined samples; the obtained plots are the same for all five examined samples. The voltage-time plot contains three main stages, which are clearly detected on the plot (Figure 1a). In the initial stage, the voltage increases rapidly up to 80 V. At this stage the amorphous layer is formed (Figure 1a, area I) and the process is accompanied by the gas bubbles appearance on the materials’ surface, which was also shown previously in the work of Snizhko et al. [22]. In the second stage of the process, the voltage has reached 105 V and dielectric breakdowns came out with the uniformly appeared electric sparks on the alloys’ surface (Figure 1a, area II). At this stage, an amorphous to crystalline structure transition of the oxide coating occurred, as described in work of Hussein et al. [23]. In the last stage of the MAO process, the voltage was near the plateau (Figure 1a, area III). Here, sparks became larger and more intense, which corresponds with the increase in the charge transfer resistance. This behavior is also confirmed by work of Mortazavi et al. [24].

On the current-time plot, the rapid drop of the current at the initial stage is detected (Figure 1a, area I). However, the process current begins at 0 A, and it extremely quickly reaches the maximum value as a result of the quasi short circuit. Simultaneously, the amorphous layer is rapidly growing. Its growth terminates as sufficient layer thickness is reached. In the second stage of the processing, the transformation of the oxide layer occurs resulting in the formation of low conductive coating (Figure 1a, area II), and finally stabilizes (Figure 1a, area III). The same behavior was also observed by Ahounbar et al. [25].

The obtained waveform that was determined by the nature of the process is shown in Figure 1b. τ_off_ refers to the period when the current is not supplied. At this period of time, titanium ions may release to electrolyte or may form a thin amorphous layer on the alloys’ surface. When the voltage value is τ_on_ positive (anodic polarization), the current is supplied, and micro discharges appear over the dielectric breakdown potential. At this period of time, the MAO process is accompanied by a strong electric field and a high temperature and pressure [6]. Thus, several components of the electrolyte, the coating, or the substrate may ionize and decompose resulting in titanium cations release. This strong electric field also affects the diffusion of the oxygen anions towards the Ti substrate. Oxygen ions interact with Ti ions, producing titanium oxide amorphous structure that later transforms into the crystalline phase. Newly formed TiO_2_ coating is subjected to the compaction during subsequent dielectric breakdowns. When the value is τ_on_ negative (cathodic polarization), the dielectric breakdowns do not occur due to the high electrical conductivity of the created oxide coating [9].

### 3.2. Surface Characterization

Coating morphology and cross-sections images of the produced surfaces by MAO processing are shown in Figure 2 and Figure 3.

Investigation of the surface morphology presented by SEM in Figure 2 revealed numerous micro-sized and sub-micron pores. These pores are formed in the process as the result of dielectric breakdowns at the extreme high temperature. Surface porosity change was also detected on the morphology images where sub-micron pores tend to disappear with the frequency increase. This was also clearly detected on the cross-section images presented in Figure 3. 

MAO treatment is a high-energy process associated with a high current and a high voltage that affect surface morphology formation. The distribution of this current and voltage over the surface is not uniform and strongly depends on the localized dielectric properties of the developed coating. Thus, appearance of pores on the surface is spontaneous during the dielectric breakdowns, which are also influenced by their size and form. Therefore, control of the coating porosity may be done by the variety of the current frequency.

Additionally, to reduce porosity, the cross-sectional images also evaluated compaction of the coating with the growth of current pulses in MAO process. Thickness of the coatings was determined on the SEM cross-sectional images. It is clearly seen that the thickness of the produced coating reduced with the current pulse increase. Sam–200 Hz has a coating thickness of 2 μm, which was reduced up to 1 μm for the Sam–1000 Hz. This can be attributed to the reduction of current peak time duration with the increase of the frequency as the result of τ_on_ and τ_off_ reduction. 

Observation of Figure 3a–c revealed almost the same coating thickness with different pores size and their number. The large pores detected in Sam–200 Hz became smaller—mostly sub-micron—in Sam–600 Hz due to the dielectric breakdowns caused by localized high temperature. This phenomenon may be attributed to the decrease of the localized temperature as the result of a specific power density reduction. With the increase of the current frequency (Figure 3d,e), the local re-melt of the initial porous coating occurs, resulting in compaction and reduction of its porosity.

Examination of the cross-sectional images showed in Figure 3 also revealed thickness of all coatings. The thickness was found to be almost the same over the surface depth of each sample.

### 3.3. Chemical and Phase Analysis

The elemental composition of the developed coatings detected by EDS is presented in Table 2.

EDS study revealed that the appearance of Ti, Al and V originated from the Ti-6Al-4V alloy and Si originated from the electrolyte decomposition. The chemical composition of all samples is almost the same and is not affected by the applied current pulse.

Figure 4 shows PIXE spectra of the formed coatings.

PIXE analysis clearly detected the following chemical elements: C, O, Al, Na, Si and Ti. Titanium and oxygen are main elements of the developed titanium oxide coating, aluminum is the major alloying element in Ti-6Al-4V alloy, and carbon peak was detected because the surface was coated in order to eliminate charge effect. Sodium and silicon were detected because of their appearance in the coating as the result of electrolyte decomposition during MAO processing. As it is clearly seen in Figure 4, there are no changes in the detected peaks were found in all examined samples.

The phase composition of the obtained coatings was revealed by the XRD measurements and are shown in Figure 5.

XRD measurement evaluated the presence of the titanium phase (ICDD 44-1294) along with two titanium oxide phases, rutile (ICDD 21-1276) and anatase (ICDD 21-1272). It is detected that the content of the rutile phase was reduced with the growth of the processing frequency. It is clearly seen on the rutile analytical peak intensity change at 27.6°, that the peak is relatively low for Sam–1000 Hz as compared to the same peak for Sam–200 Hz alloy. Additional peaks of rutile at 36.2 and 41.2° were also clearly detected in the pattern obtained in the Sam–200 Hz alloy, and their intensity was significantly reduced with the current frequency increase. Simultaneously, the analytical peak of anatase at 25.5° and its additional peak at 55.5° intensity were detected to be increased with the current frequency increase. This behavior may be attributed to the anatase-to-rutile transformation that occurred when the process reached a dielectric breakdown temperature, as described in by Hanaor et al. [26]. A metastable anatase phase is irreversible and transforms into rutile phase at a low frequency where the contribution of the current is more significant. No more phases were detected by XRD analysis. However, additional phases originated from the electrolyte decomposition may appear in the coating in amorphous structure or even in crystalline with a content that is below the detection limit of XRD.

Apart from XRD analysis, the TiO_2_ coatings obtained by MAO process were also investigated by Raman spectroscopy (Figure 6).

Raman spectra illustrated in Figure 6 are almost the same for all examined samples and show that the TiO_2_–anatase phase, which was detected with the active vibrations at: 147 (Eg1), 198 (Eg2), 398 (B1g), 515 (A1g + B1g doublet band), 640 (Eg3), and 796 cm^−1^ (B1g[F], first overtone of B1g at 398 cm^−1^) [27]. The intensity of the Raman spectra peaks increased with the current frequency increase as the result of the produced surface porosity variation. 

Due to the low penetration, the rutile phase was not detected by Raman spectroscopy. However, the anatase-to-rutile phase transformation was found by XRD and described above. Consequently, rutile was formed as the result of the transformation in the inner layer of the coating.

### 3.4. Corrosion Resistance Investigation 

The corrosion properties of the treated samples were determined by a potentiodynamic polarization method. The obtained curves are illustrated in Figure 7.

According to theory, the higher the corrosion potential, the higher the corrosion resistance. Potentiodynamic polarization curves on Figure 7a clearly indicated a shift of the coated samples to the more positive potential. This may be attributed to the reduction of the anodic and the cathodic processes due to the oxide coating appearance.

The polarization resistance (R_p_) was calculated according to Equation (1):(1)$$R_{p}=\frac{\beta_{a}\times \beta_{c}}{2.3\times i_{\mathrm{corr}}\left(\beta_{a}+\beta_{c}\right)}.$$

The Tafel slopes β_a_ and β_c_ were calculated from the anodic and the cathodic curves on the plot (Figure 7b). The corrosion potentials (E_corr_), the corrosion current densities (i_corr_) and the polarization resistance (R_p_) are listed in Table 3.

Calculation results in Table 3 revealed that the corrosion rate of coated alloy was at least 40 times lower compared to the untreated alloy. The lowest value was obtained on Sam–1000 Hz, which exhibited 125 times higher corrosion resistance in comparison with untreated alloy. Thus, the rate of the corrosion was 0.000150 mm/year and 0.018644 mm/year for the Sam–1000 Hz and untreated alloy, respectively. Furthermore, the higher the applied current pulse in MAO process, the higher the corrosion resistance. It is well correlated with the morphology investigation of the coatings where Sam–1000 Hz was found to be more compact with lower porosity compared to other examined samples.

## 4. Conclusions

In the current work, ceramic oxide coating on the Ti-6Al-4V alloy using MAO processing has been produced. The effect of different current pulse frequency on the coating morphology, phase composition and corrosion resistance has been studied. It was found that the application of different current frequency affected the porosity and compaction of the formed oxide coating.

Surface morphology and cross-sectional investigation by electron microscopy showed that the higher the applied current pulse frequency, the more compact and less porous the obtained coating is. After processing with a pulse current of 1000 Hz, the coating with a thickness of 1 µm and with a low porosity has been obtained. PIXE analysis revealed the appearance of titanium oxide coating on treated alloys. Two phases of titanium oxide, anatase and rutile were identified by XRD. Raman spectroscopy investigation detected the appearance of anatase phase on the surface of the examined samples. Rutile appeared in inner layer of the coating as the result of the partial transformation of the anatase phase.

Potentiodynamic polarization curves have evaluated that the corrosion resistance of the produced coating 40 times higher compared to the untreated alloy. Moreover, it was found that the higher the current pulse frequency, the higher the corrosion resistance of the obtained coating resulting in the formation of compacted and low porous oxide coating.

## Figures and Tables

**Figure 1 materials-12-03983-f001:**
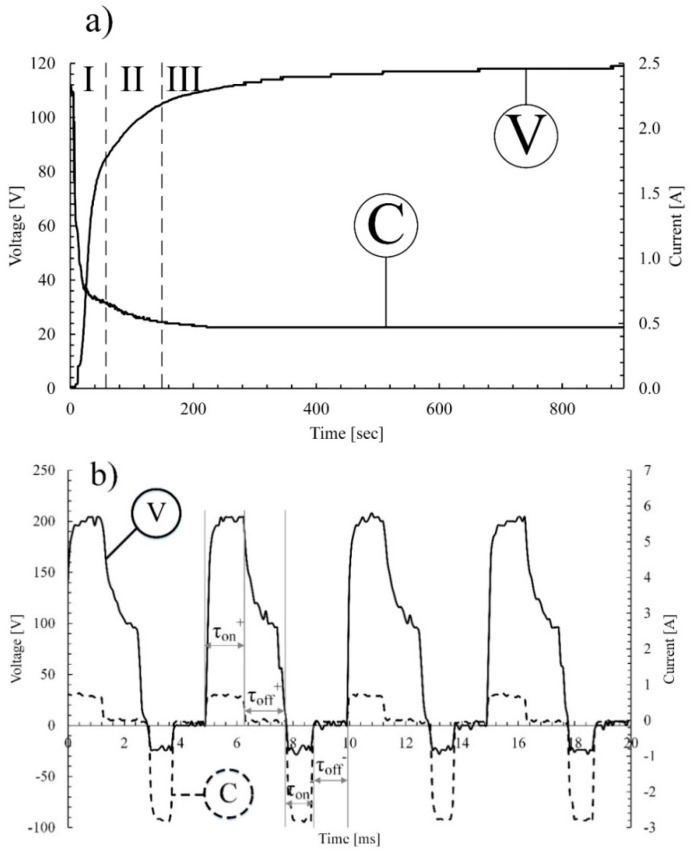
Typical plots of: (**a**) voltage- and current-time behavior; (**b**) the waveform of the MAO process applied on alloy Ti-6Al-4V. V—voltage-time curve; C—current-time curve.

**Figure 2 materials-12-03983-f002:**
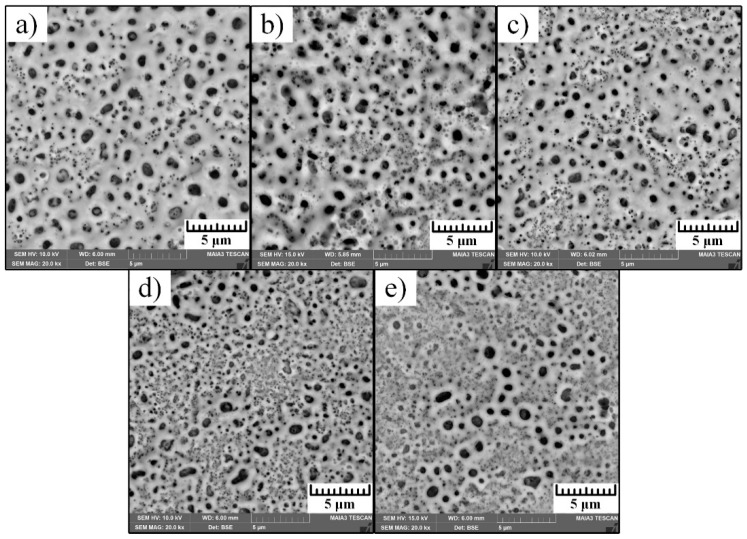
SEM images of surface morphology of the alloy Ti-6Al-4V treated by MAO with various current pulses: (**a**) Sam–200 Hz; (**b**) Sam–400 Hz; (**c**) Sam–600 Hz; (**d**) Sam–800 Hz; (**e**) Sam–1000 Hz.

**Figure 3 materials-12-03983-f003:**
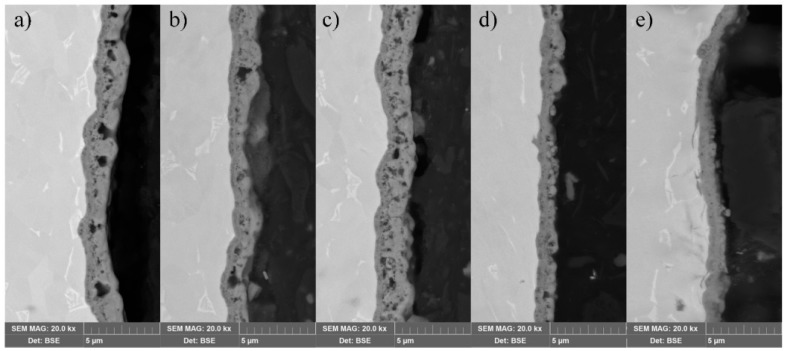
SEM images of cross-sections of the alloy Ti-6Al-4V treated by MAO with various current pulses: (**a**) Sam–200 Hz; (**b**) Sam–400 Hz; (**c**) Sam–600 Hz; (**d**) Sam–800 Hz; (**e**) Sam–1000 Hz.

**Figure 4 materials-12-03983-f004:**
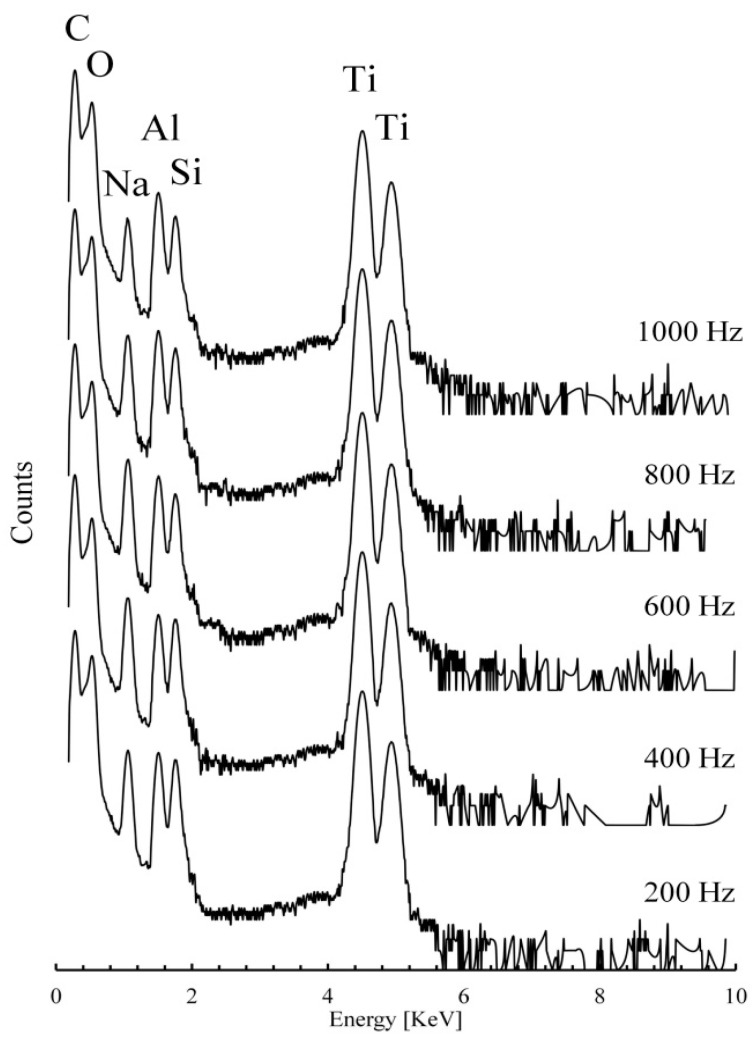
PIXE spectra of the alloy Ti-6Al-4V coatings subjected to MAO treatment with various current pulses.

**Figure 5 materials-12-03983-f005:**
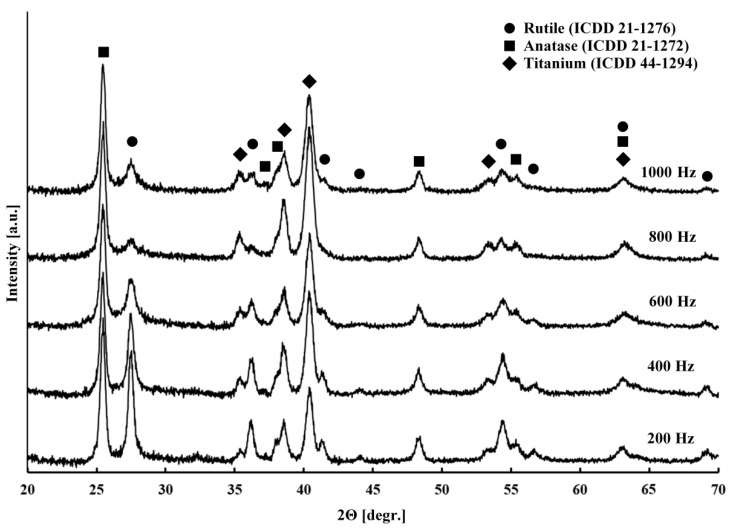
X-ray diffraction patterns of the alloy Ti-6Al-4V surfaces after MAO treatment with various current pulses.

**Figure 6 materials-12-03983-f006:**
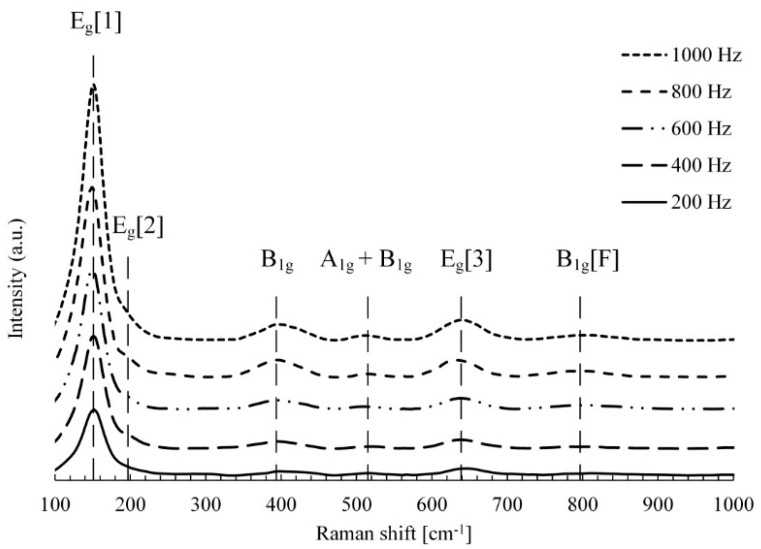
Raman spectra of the alloy Ti-6Al-4V surfaces after MAO treatment with various current pulses.

**Figure 7 materials-12-03983-f007:**
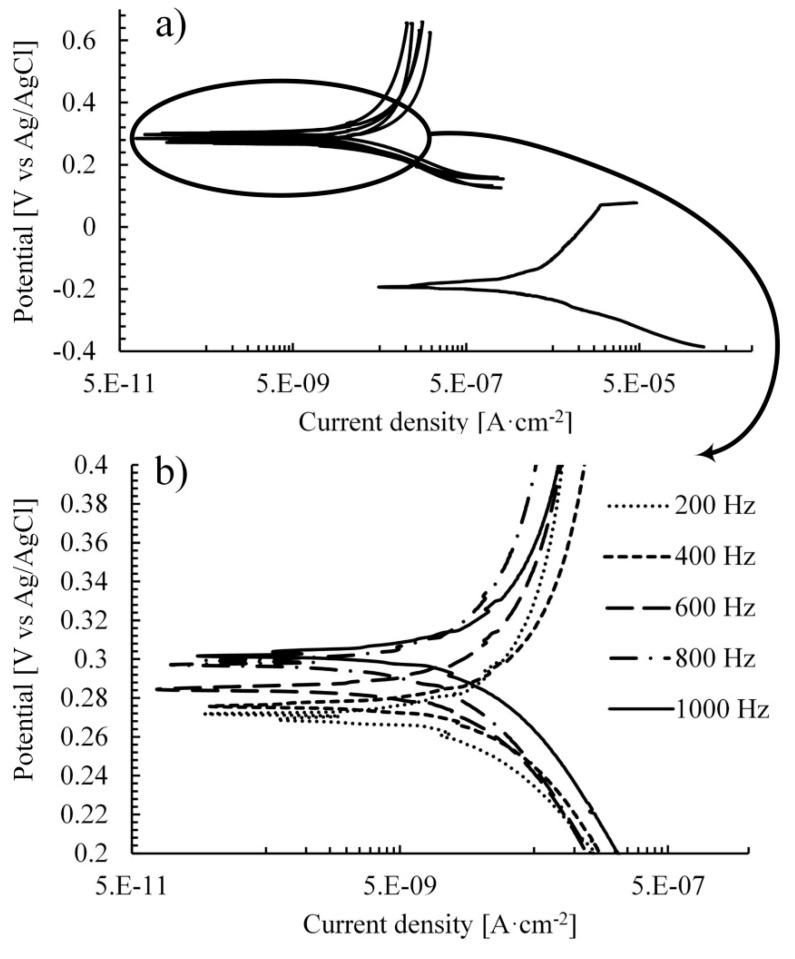
(**a**). potentiodynamic polarization curves for alloy Ti-6Al-4V after MAO treatment with various current pulses (upper curves) and curve for base Ti-6Al-4V (lower curve). (**b**). enlarged curves for treated alloys. Examination was carried out in 3.5 wt.% NaCl.

**Table 1 materials-12-03983-t001:** Names of examined samples in the research work.

Name of the Sample	Frequency Applied (Hz)
Sam–200 Hz	200 Hz
Sam–400 Hz	400 Hz
Sam–600 Hz	600 Hz
Sam–800 Hz	800 Hz
Sam–1000 Hz	1000 Hz

**Table 2 materials-12-03983-t002:** Elemental composition of the alloy Ti-6Al-4V treated by MAO with various current pulses detected by EDS investigation (at. %).

Sample	Ti	Al	V	Si	O
Sam–200 Hz	24.2 ± 1.3	2.3 ± 0.5	0.9 ± 0.1	5.1 ± 0.9	Balance
Sam–400 Hz	24.1 ± 1.5	2.5 ± 0.3	0.9 ± 0.1	3.5 ± 0.7	Balance
Sam–600 Hz	24.4 ± 2.1	2.8 ± 0.3	1.0 ± 0.2	3.7 ± 0.7	Balance
Sam–800 Hz	27.4 ± 2.3	3.1 ± 0.7	1.1 ± 0.3	2.2 ± 0.6	Balance
Sam–1000Hz	24.0 ± 1.2	2.5 ± 0.5	0.9 ± 0.2	4.4 ± 1.0	Balance

**Table 3 materials-12-03983-t003:** Calculated corrosion test results for alloy Ti-6Al-4V after MAO treatment with various current pulses and the base alloy.

Samples	E_corr_ (mV)	i_corr_ (nA/cm^2^)	β_a_ (mV/decade)	β_c_ (mV/decade)	R_p_ (kΩ cm^2^)	Corrosion Rate (mm/year)
Untreated alloy	−198.386	2141.5	310.368	88.89	14	0.018644
Sam–200Hz	270.387	49.319	333.266	130.897	832	0.000429
Sam–400Hz	269.917	33.000	304.585	92.802	937	0.000287
Sam–600Hz	303.337	30.534	225.418	117.102	1097	0.000266
Sam–800Hz	282.228	33.364	308.202	129.618	1189	0.000290
Sam–1000Hz	300.085	17.260	216.575	113.596	1877	0.000150

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
