# Peer review of "Study of the Effect of Current Pulse Frequency on Ti-6Al-4V Alloy Coating Formation by Micro Arc Oxidation"

_materials, 2019, doi:10.3390/ma12233983_

Round 1
Reviewer 1 Report
Check the title: ”Study of The Effect of Current Pulse on Ti Coating Formation by Micro Arc Oxidation”. From the title results that are coating with titanium … which is not in the paper. The paper describe coating of an Ti-based alloy!! Why you choose Na2CO3 and Na2SiO3 for titanium alloys? Why is it important? Why didn’t use others? For what applications this covered alloy is suitable? Please insert in text. Compared to the classic alloy (Ti6Al4V), how much is the corrosion resistance improved? please make a comparison in the text.Author Response
Dear Reviewer,
Attached file is point-by-point response to your comments.

Reviewer 2 Report
see attached file

Author Response
Dear Reviewer,
Attached file is point-by-point response to your comments.

Reviewer 3 Report
This article is of scientific interest for the engineering of new coatings by a promising micro-arc oxidation method on the surface of titanium alloys, providing favorable functional properties of the alloy surface. However, the article requires considerable revision, including improvement of English grammar, correcting the title of the article, and adding a discussion of the scientific results in comparison with the results obtained by other authors.
Following comments:
The title of article should be corrected. Because, the studies were focused on effect of only “Current Pulse Frequency” on the “coating formation on Ti-6Al-4V alloy” In the Abstract, authors wrote about Ti alloy. Although, only specific Ti6Al4V alloy was investigated. Therefore, it is not clear what materials did you mean in cases of “This alloy…” (13 line) and “…base Ti alloy…” (14 line). From the Introduction, it is not clear for what application area micro-arc coatings were created. Since the application field determines what the most important property of the formed coatings (e.g., morphology, structure, composition, corrosion resistance, mechanical properties or etc.) are required. Authors should extend the Introduction with this description. Besides, in the Introduction, the authors indicate that in addition to the electrolyte composition, the electrophysical parameters of MAO also affect the properties of the formed coatings. However, authors do not give any references to published works on this subject and do not give information how the parameters of the MAO process can affect the coating properties. It was not seen the MAO time for the coating deposition in “Materials and Methods” Section. The pauses in the second parts of both semi-periods (when the current is not supplied), and differences between height and form of anodic and cathodic pulses of voltage and current are observed in Fig 1 (b). However, the description of these events was not provided in the “Materials and Methods” Section? How do you explain the fact that the height of the voltage anodic pulses in Fig. 1 (b) is equaled to 200 V that below applied voltage of 250 V (59 line). Where is the discussion in Section 3.2 how the current pulse frequency effect on the size and distribution of the pores on the surface and in the cross-section of the coatings? It is well evident in Fig. 2 that with increasing of the pulse frequency that not only sub-micron pores appears, but also the micro-sized pores are disturbed non-homogeneously on the coating surface. Why is it? Why is “the cross-sectional images evaluated also compaction of the coating with the growth of current pulses in MAO process” (137 line)? The scale is not clearly seen in Fig. 2. It is clearly seen that the Sam-600Hz coating (Fig. 3 c) is the thickest or has a thickness like that of Sam-200Hz coating (Fig. 3a). Why did the authors not explain this? Where the proof of following thesis is (143 line): “The produced coatings are also found to be homogeneous in the thickness along the Ti substrate”? Why were Na and С elements not detected in the coatings by EDS? In addition, the authors should provide the measurement error in Table 2 to understand whether the differences in values are not statistically significant. Why was V element not detected in the coatings by PIXE? The description of XRD results is not enough for full understanding. As evident in XRD patterns, with decreasing of the current frequency the intensity of reflections from rutile phase increase (not only peak at 27.44 ° but also peaks at 36.08 °, 54.32 ° etc.), but intensity of reflections from anatase phase does not change. Where the experimental evidence of following thesis is: “A metastable anatase phase is irreversible transforms into rutile phase at a low frequency where the contribution of the current is more significant.” (174-175 lines). From Raman spectra, the authors concluded that: “…anatase was partially transformed to a rutile phase when the process was reached a dielectric breakdown temperature” (187-188 lines). Oppositely, it is well evident from Raman spectra that with increasing of the current frequency the intensity of lines from the active vibrations of anatase phase in the coatings increased. To Section 3.4 and Conclusions, the same remark as was in the Abstract. You wrote: sometimes “…base Ti…” (Table 3), sometimes “…base alloy” (212 line) or “…base Ti alloy” (230 line). Did you mean different materials or only Ti6Al4V alloy? Please be careful with the terminology as it is difficult to comprehend.
Author Response

(The authors gave the same response as above.)

Round 2
Reviewer 1 Report
The paper was significantly improved and can be published!
Author Response
Thank you very much.
Reviewer 3 Report
Following remarks:
In section 2.1 you added: “The final roughness of the surface was Ra = 2.5.”. What units? If you mean microns, it is impossible. After polishing with abrasive paper grits up to P1200, the average roughness (Ra) of Ti and its alloys should not exceed 0.5 µm. The average roughness of 2.5 µm is too high value a greater than the thickness of the formed coatings. In section 3.3 you wrote: “PIXE analysis clearly detected the following chemical elements in the formed coating: C, O, Al, Na, Si and Ti.” However, previously in response you assured that carbon was not presented in examined samples. Please, resolve this contradiction. Please, provide a more detailed XRD analysis in section 3.3 explaining the phase transformation from anatase to rutile. Presented description of changing of single reflection at 27.6 degrees in XRD patterns does not give a complete picture of the phase transformation. The previous remark “From Raman spectra, the authors concluded that: “…anatase was partially transformed to a rutile phase when the process was reached a dielectric breakdown temperature” (187-188 lines). Oppositely, it is well evident from Raman spectra that with increasing of the current frequency the intensity of lines from the active vibrations of anatase phase in the coatings increased.” is still. Please, provide a qualitative analysis why the intensity of the peaks corresponding anatase phase vibrations increases with the increasing of the current frequency.
Author Response
Dear Reviewer,
Please see attached answers to your comments.
